# QuantWear: Quantum-Scale Wear Particle Detection for Jet Engine Diagnosis

**Zheng Wang** [1,2]  **Yanwen Wang** [1]  **Tianyu Fang** [3]  **Jiaxing Shen** [4]  **Yisen Kang** [5]  **Di Wu** [1]  **Yuanqing Zheng** [6]

## Abstract

The quantity and 3-D shape of wear particles are essential indicators for assessing the health of jet engines, enabling early detection of potential damage and preventing accidents caused by catastrophic failures. However, capturing wear particles is difficult due to their minute sizes and ultra high-speed movement within intense jet flows. Existing technologies struggle with the extreme background noise and low resolution in such harsh environments. In this paper, we propose QuantWear, the first quantum sensing system designed to directly quantify and profile wear particles on the sub-millimeter scale. QuantWear innovatively tracks wear particles by monitoring the spectral signatures of Sodium (Na) and Potassium (K) atoms within jet flow, which naturally adhere to particle surfaces due to electrochemical reactions in high-temperature combustion. We construct a custom atomic detector that leverages quantum jump and Faraday rotation effects to isolate these specific atomic signals, effectively suppressing the broad-spectrum flame noise. Next, we apply a deep learning framework to effectively measure the quantity of wear particles in dynamic vaporous backgrounds. Finally, we generate a fully reconstructed 3-D model of the wear particles from multiple 2-D images. Extensive field tests and high-fidelity simulations demonstrate that QuantWear achieves an imaging Signal-to-Noise Ratio (SNR) of 22.5 dB and a 3-D reconstruction similarity exceeding $95\%$, significantly outperforming state-of-the-art technologies.

[1]Hunan University, Changsha, China [2]China Electric Power Research Institute, China [3]University of Texas at Dallas, Richardson, TX, USA [4]Lingnan University, Hong Kong SAR, China [5]Central South University, Changsha, China [6]The Hong Kong Polytechnic University, Hong Kong SAR, China. Correspondence to: Yanwen Wang <wangyw@hnu.edu.cn>, Di Wu <dwu@hnu.edu.cn>.

*Proceedings of the 43rd International Conference on Machine Learning*, Seoul, South Korea. PMLR 306, 2026. Copyright 2026 by the author(s).

## 1. Introduction

### 1.1. Motivation

Jet engines serve as the primary power source for commercial aviation, military aircraft, missiles, spacecraft, and turbines, playing a crucial role in aerospace, military, space exploration, and energy conservation. However, engine malfunctions can incur severe accidents, including engine explosions. As such, health monitoring of jet engines has become increasingly essential, facilitating early diagnosis of potential damage and extending their operational life. When a jet engine operates normally, extremely high temperatures in the combustion chamber and elevated pressures in the compressor generate thermal expansion and friction among engine components. Meanwhile, the turbine and compressor blades are susceptible to erosion from high-temperature gases and unburned fuel. Anomalies in the cooling system further exacerbate engine wear and tear, resulting in the presence of wear particles (refer to details in Appendix A.1) expelled along with jet flows (Hadavi et al., 2015). Both the quantity and shape of wear particles aid in determining the operational conditions, rendering them critical indicators for effective jet engine health monitoring.

### 1.2. Prior Works and Limitations

Existing technologies struggle with the extreme background noise and low resolution in such harsh environments, making it difficult to accurately measure tiny, ultra high-speed objects in complex environments. Existing works to sense the wear particles can be roughly divided into three categories, electrical parameter-based, vision-based, and optical-based. Electrical parameter-based approaches leverage the fact that electrical parameters, such as current, capacitance, and inductance, can be impacted by the jetted wear particles, which can be measured by corresponding sensors (Muthuvel et al., 2018; 2021). However, electrical parameter-based methods are only available for detecting the presence of wear particles in normal environments (Muthuvel et al., 2021). Traditional studies use cameras to capture and analyze wear particles, enabling remote data acquisition without damaging the devices and offering visualization for analysis (Liu et al., 2023; You et al., 2024; Daugherty et al., 2025). However, high sensitivity to ambient light significantly degrades the quality of images, causing motion blur when capturing high-speed objects. Optical sensor-based

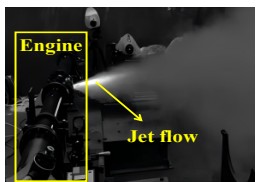

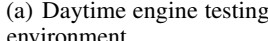

(a) Daytime engine testing environment.

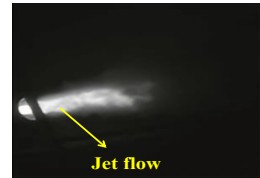

(b) Nighttime engine testing environment.

*Figure 1.* Engine which high-speed and intense jet flow.

methods capture wear particles with high precision and use non-contact measurement. However, current optical sensors lack the sensitivity and dynamic range of light required to differentiate high-speed wear particles from bright backgrounds (Martín et al., 2020; Shah et al., 2024).

### 1.3. Challenges and Solutions

The fundamental challenge lies in capturing minute wear particles within high-speed, intense jet flows and high-brightness backgrounds, as in Figure 1, which exceeds the capabilities of traditional technologies. To address this, we develop a quantum optical system coupled with a electro-optical imaging system, specifically tailored for Sodium (Na) and Potassium (K) atoms. These elements, common in combustion catalysts, form elemental Na and K[1] that adhere to wear particles due to electrochemical reactions in high temperatures. By leveraging quantum jump and Faraday rotation effects, our system acts as an ultra-narrowband filter, isolating the spectral signatures of adhered Na and K atoms while completely suppressing the broad-spectrum flame noise, effectively rendering the background 'invisible'. As such, wear particles can be successfully filmed by the electro-optical imaging system.

Second, accurately quantifying wear particles is intricate due to their tiny and diverse sizes, high-speed movement and unpredictable trajectories in dynamic flows. We employ the YOLOv8 network, which is particularly excels in detecting small objects in cluttered environments due to its unique architecture and advanced feature extraction capabilities. Unlike traditional tracking methods (e.g., Kalman Filter) that struggle with time-varying vaporous backgrounds, YOLOv8's multi-scale architecture and advanced suppression algorithms enable robust tracking, counting, and labeling of high-speed wear particles.

The third challenge involves accurately measuring the 3-D geometric information of wear particles as our quantum images are inherently 2-D spectral projections lacking depth information. To resolve this, we treat the temporal sequence of 2-D silhouettes captured from a single fixed camera as inputs from virtual multi-viewpoints, reconstructing them into 3-D representations to estimate their shapes. Given

---

[1]An elementary substance is a chemical substance whose molecules are only formed out of one kind of atoms.

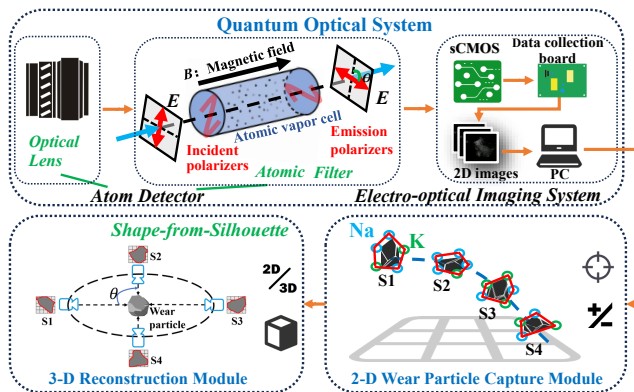

*Figure 2.* System overview.

that wear particles in jet flows rotate rapidly, our system captures diverse angular perspectives over time. We employ the Shape-from-Silhouette (SFS) approach to synthesize these multi-angle contours into comprehensive 3-D shapes.

Our main contributions are summarized as follows:

(1) We develop a quantum optical system capable of efficiently capturing wear particles in high-speed jet flows. To the best of our knowledge, we are the first to monitor wear particles in jet engines using quantum sensing techniques.

(2) We harness the spectral and chemical properties of Na and K atoms and incorporate a deep learning framework for assessing the quantity and size of wear particles effectively.

(3) We achieve a 3-D representation of wear particles from multiple 2-D images, offering precise information about the 3-D shape of wear particles.

(4) We implement our QuantWear system in field-test environments across various types of jet engines. QuantWear can accurately measure the quantity and 3-D shape of wear particles present in jet flows.

## 2. System Overview

The overall design of QuantWear is illustrated in Figure 2. QuantWear consists of a Quantum Optical System, a 2-D Wear Particle Capture Module and a 3-D Reconstruction Module. The quantum optical system employs an optical lens with a wide field of view (FoV) to fully capture the light generated by engine jet flows. The captured light is then filtered at the atomic level by a synthetically designed atomic filter to extract the desired light with a wavelength specifically matching the spectrum of Na and K. The filtered light is inputted into an electro-optical imaging system to generate 2D images. The 2-D Wear Particle Capture Module then employs modified YOLOv8 to accurately measure the quantity and 2-D shapes of wear particles by pinpointing the elemental Na and K in the images. Finally, the 3-D Reconstruction Module leverages the rich silhouette information of wear particles in 2-D images filmed from diverse shooting angles, enabling a comprehensive reconstruction from an entirely 3-D perspective.

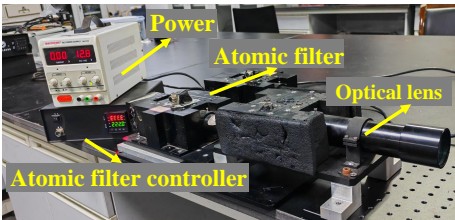

*Figure 3.* Hardware of Atom Detector.

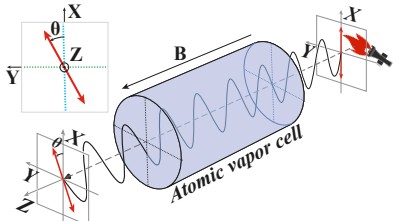

*Figure 4.* Faraday rotation of the light.

# 3. System Design

## 3.1. Quantum Optical System Design

Detecting sub-millimeter, high-velocity ($> 100$ m/s) wear particles in jet exhaust presents a critical optical challenge due to intense, broad-spectrum blackbody radiation. This background noise saturates standard sensors, yielding near-zero SNR. To overcome this, we designed a quantum optical system functioning as an ultra-narrowband filter. By isolating the spectral signatures of Na and K, alkali metals inherent in combustion that adhere to wear particles, the system effectively renders the flame transparent.

We propose a synthetic design for an atomic-level optical imaging system tailored for rapid jet flows. The architecture (Figure 2) maximizes photon collection efficiency while enforcing strict spectral selectivity. The system integrates an atom detector and an electro-optical imaging unit with a high-speed camera. The atom detector exclusively transmits the spectral lines of Na and K, filtering out all other wavelengths. Subsequently, the imaging system, utilizing a super-fast shutter, captures the elemental Na and K adhering to the surfaces of the high-speed wear particles.

The image-formation process contains four steps. First, the optical lens collects broad-spectrum emission from the jet flow, including Na and K signatures attached to wear particles. Second, the light enters the heated vapor cell, where gaseous Na/K atoms selectively interact with photons matching their atomic transition energies and re-emit resonant photons through atomic transitions. Third, the external magnetic field induces Faraday rotation for the resonant Na/K component passing through the atomic medium. Finally, the output polarizer blocks the unrotated, broad-spectrum background flame while transmitting only the rotated Na/K light. As a result, the electro-optical imaging system forms high-contrast 2-D images dominated by Na/K signals associated with wear particles.

### 3.1.1. ATOM DETECTOR

**Optical lens and atomic filter design.** The atom detector integrates an optical lens and an atomic filter (Figure 2). The lens collects jet-flow emission after atmospheric absorption and dispersion. Refer to detailed in Figure 17 in Appendix, this lens provides a Field of View (FoV) sufficient to track wear particle trajectories. Crucially, the atomic filter acts as

an ultra-narrowband selector, exclusively passing the spectral lines of Na and K. Finally, the electro-optical imaging system converts this filtered signal into a 2-D image.

**Atomic filter operation and quantum jump.** The atomic filter is the core component governing spectral transmission and image contrast. It consists of a heated atomic vapor cell, a temperature controller, a magnetic-field generator, and customized polarizers. Comprising an atomic vapor cell, temperature controller, and magnetic field generator, it functions as an optical bandpass filter. By leveraging nonlinear frequency conversion within the cell, the system achieves precise spectral isolation, transmitting target wavelengths while effectively rejecting off-band signals.

However, unlike simple bandpass filtering, optical selection is more complex. In our design, this filtering process is achieved within a developed quartz atomic vapor cell heated by a temperature controller to vaporize the enclosed Na or K. Within an external magnetic field, the gaseous atoms absorb incident photons matching their transition levels and re-emit them via quantum jumps (refer to detials in Appendix A.2). By tuning the cell temperature, we control the atomic density and, consequently, the photon emission intensity. The hardware configuration is depicted in Figure 3.

**Magnetic field and Faraday rotation control.** To further facilitate precise control over light typically meeting the spectra of Na and K, a magnetic field generator is employed to create an external magnetic field covering the vapor cell. This magnetic field induces Faraday rotation in light near the atomic resonance frequencies of Na and K, resulting in alteration of their polarization, while off-resonance light remains unaffected, as illustrated in Figure 4 (refer to details of Faraday rotation in Appendix A.3). We precisely configure the atomic vapor cell's temperature and magnetic field intensity to modulate polarization rotation for effective light filtration. Permanent magnets are employed for field generation since unlike electromagnets, they offer superior stability and a lightweight volume, making them ideal for the atomic filter's compact, high-performance design.

### 3.1.2. HIGH-SPEED ELECTRO-OPTICAL IMAGING SYSTEM

In our work, we develop an electro-optical imaging system based on an optical setup that includes a scientific CMOS

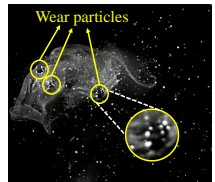

(a) Wear particles at frame 1.

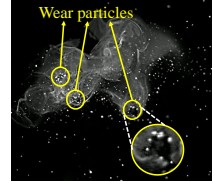

(b) Wear particles at frame 2.

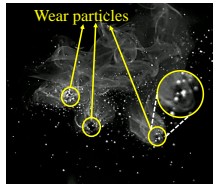

(c) Wear particles at frame 3.

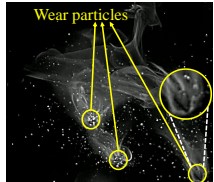

(d) Wear particles at frame 4.

*Figure 5.* Wear particles captured at continuous frames.

*Table 1.* High-speed camera parameter configuration

| Parameter | Configuration |
|---|---|
| Resolution | Full resolution $1280 \times 800$ |
| Max Frame Rate | 2200 fps |
| Pixel Size | 25.6 μm |
| CMOS Chip Size | $28.16mm \times 19.71mm$ |
| Image Bit Depth | 8bit |

(sCMOS) detector module and a high-speed data collection board to detect and process the collected images. Compared to the traditional approach that directly connects the camera to the computer via a Gigabit Ethernet interface, our solution significantly enhances image transmission speed with this data collection board, effectively supporting the capture of high-speed moving targets.

Our system utilizes an sCMOS sensor to convert filtered optical signals into electrical data, delivering low noise, high frame rates, and superior Quantum Efficiency (QE). As a measure of photoelectric conversion efficiency, QE represents the ratio of collected charge carriers to incident photons at a specific wavelength (de Mello et al., 1997). A high-speed data collection board packages these signals for transmission to the host PC via PCIe. Conversely, system initialization and camera parameter configuration, as shown in Table 1, are controlled by the host through an RS422 interface. Refer to detailed validation of our developed quantum optical system in Appendix B.

### 3.2. Capture Wear Particles

#### 3.2.1. PINPOINT WEAR PARTICLES IN 2-D IMAGES.

The next step is to precisely locate the wear particles in the acquired 2-D images. Note that this is a non-trivial task since our developed quantum optical system, while capable of precisely capturing Na and K atoms, filters out all other light spectra, including the wear particles themselves. In other words, except for the light within the spectra of Na and K atoms, all light is intentionally annihilated, resulting in a dark background (refer to Figure 18(b) in Appendix B).

To solve this problem, we leverage the physicochemical properties of Na and K, which react electrochemically with aluminum oxide (i.e., $Al_2O_3$, the primary component of wear particles (Bensalah et al., 2009)). Driven by low ionization energy, Na and K transfer electrons to $Al_2O_3$ particu-

larly in high-temperature environments, inducing strong adhesion and synchronized movement. Conversely, unbound atoms disperse randomly within the jet flow. Therefore, mapping the spatial distribution of Na and K serves as a beacon for tracking the wear particles.

Figure 5 depicts moving wear particles identified through Na and K detection. We observe three particles surrounded by dense Na/K clusters representing their trajectories. Conversely, unattached elemental Na and K in the jet flow exhibit a dispersed, non-compact pattern. This because at high temperature, Na/K transfer electrons to the aluminum oxide in wear particles, causing strong adhesion. Consequently, bound Na/K forms dense clusters with clear trajectories, unlike unbound combustion products that move randomly and dispersedly. This difference in spatial distribution enables accurate isolation of wear particles from the background.

#### 3.2.2. WEAR PARTICLES NUMERATION

However, except for elemental Na and K, the images display less bright and highly variable vaporous patterns in the background. This interference arises from broadband flame radiation that penetrates the atomic filters, which cannot be adequately filtered by the atomic filter. Therefore, the time-varying nature of this background noise renders conventional algorithms ineffective for accurate particle localization. In our work, we modifies YOLOv8's architecture into a hybrid CNN-Transformer by replacing the final C2f stage of YOLOv8's backbone with a custom TR-C2f module, as shown in Figure 6. This module integrates a Transformer Bottleneck Block featuring a Multi-Head Self-Attention (MHSA) mechanism and a Feed-Forward Network. This design offers two main benefits: (1) The MHSA's global receptive field analyzes spatial correlations across the entire frame, helping to distinguish micro-particles from transient artifacts; and (2) Positioning the TR-C2f at the end of the backbone preserves high-level semantic features, boosting detection stability under extreme dynamic conditions.

Note that the standard Non-Maximum Suppression (NMS) in YOLOv8 can lead to false negatives in highly dense clusters of sub-millimeter particles. However, wear particles moving and rotating rapidly within the jet flow make severe occlusions or highly overlapping clusters statisti-

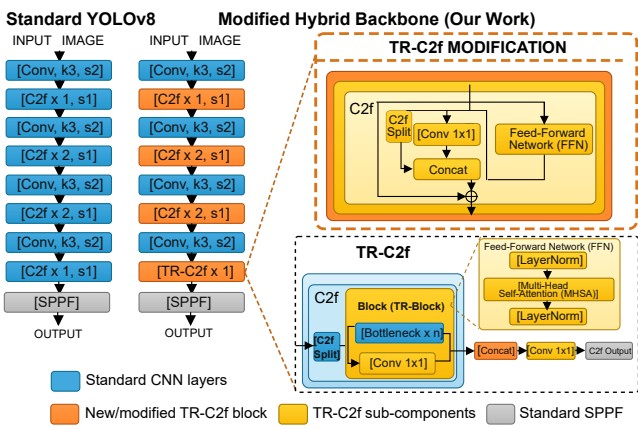

Figure 6. The architecture of modeified YOLO v8 .

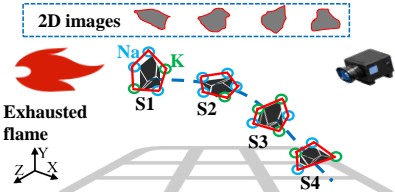

(a) Wear particles captured in different shooting angles.

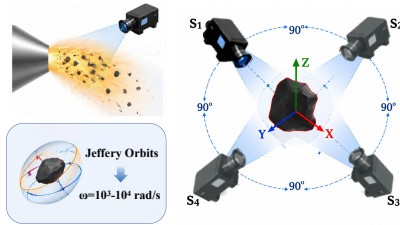

(b) Virtual cameras for capturing wear particles.
Figure 7. Capture different facets of wear particles.

cally rare. Therefore, we design a simple yet effective pre-processing algorithm to automatically remove rare overlapping instances from our training dataset before training the network. Frames are excluded if the Intersection over Union (IoU) between bounding boxes exceeds a strict threshold or if a continuous contour area significantly exceeds the average particle size. This process removed only fewer than 3000 frames ($< 0.1\%$) from our $3.168M$ frames dataset.

### 3.3. 3D Reconstruction for Wear Particles

While we have successfully characterized wear particles in jet flows, our quantum optical system is limited to planar imaging, lacking depth resolution. However, obtaining 3-D profiles for wear particles is critical, as their geometric information are essential for precise engine diagnostics.

In our study, we implement a single camera 3-D reconstruction technique that exploits intrinsic particle rotation to generate multi-viewpoint data. Unlike multiple camera arrays, this method synthesizes temporal sequences of distinct 2-D projections, as shown in Figure 7(a), to resolve surface geometry. Therefore, a comprehensive 3-D representation can be achieved given sufficient angular diversity in the captured 2-D profiles.

Specifically, we utilize the Shape-from-Silhouette (SFS) method (Liu et al., 2024) to derive 3-D particle geometry from 2-D profiles. SFS constructs the object's visual shape by intersecting silhouettes from multiple viewpoints. Validated in computer vision and medical imaging for recovering complex topology from sparse projections (Melas-Kyriazi et al., 2024), this technique is suitable for high-fidelity wear particle reconstruction.

In our scenario, a single camera remains stationary while the wear particles move and rotate at high speed due to the jet flow. Following the *Jeffery Orbits* model, extreme shear rates in high-speed jet flows induce particle angular velocities reaching thousands of rad/s (Abtahi & Elfring, 2019),

naturally exposing sufficient facets of the irregular particles for 3D synthesis from 2D inputs. To facilitate a better understanding of 3-D reconstruction, we simplify our scenario, as shown in Figure 7(b). Assume that an irregularly shaped wear particle completes a full circular rotation in one second at a constant speed, while the camera captures four frames per second. This configuration is kinematically equivalent to a *virtual* camera orbiting a stationary object. Specifically, four silhouette images of the wear particle can be obtained as if they were filmed from four virtually positioned cameras at different angles relative to the wear particle.

As such, the primary criterion for selecting the SFS method relies on our unique physical constraint: rapid particle rotation within the jet flow allows a single stationary camera to capture diverse angles over time, which provides virtual multi-viewpoints for 3-D reconstruction, making SFS highly suitable for synthesizing these projections into a 3-D bounding volume. While SFS struggles with concavities, this does not impact wear severity calculations required for assessing engine health, which primarily rely on overall volume and external shape rather than microscopic details (Kim et al., 2013).

#### 3.3.1. SILHOUETTE GENERATION OF WEAR PARTICLES

Generating silhouettes is an essential step in the SFS method, requiring the conversion of raw images into binary masks that isolate the particle foreground (i.e., the object to be reconstructed). To enhance the visibility of the silhouettes, we apply denoising and contrast enhancement to maximize separation of object and the background, as demonstrated in Figure 8(a) and (b). Then, particle boundaries are precisely delineated using edge detection operators such as Canny (Chen et al., 2024) or Sobel(Cetinkaya et al., 2024).

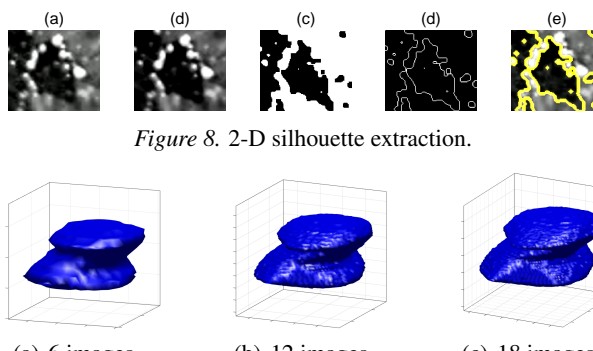

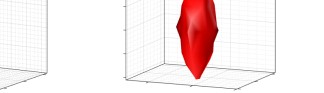

*Figure 8.* 2-D silhouette extraction.

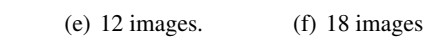

*Figure 9.* 3-D reconstruction.

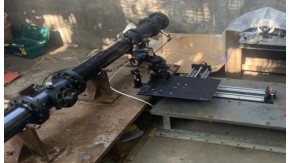 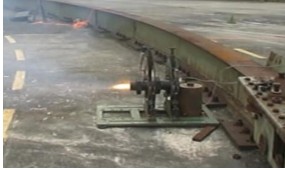

(a) Test engine 1.          (b) Test engine 2.

*Figure 10.* Part of test engines.

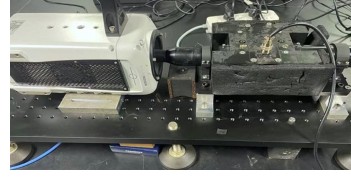

*Figure 11.* Connecting lens and high-speed camera.

After preprocessing the images, we employ binarization for foreground segmentation. Morphological operators (erosion, dilation, opening, closing) then refine the masks by eliminating noise and bridging discontinuities, as shown in Figure 8(c). Next, edge segments are linked to form closed silhouettes (Figure 8(d, e)), a process efficiently handled by standard libraries like OpenCV. Finally, we transform the silhouette coordinates using camera parameters to spatially register the multi-view data for 3D reconstruction.

### 3.3.2. VOLUME CONSTRUCTION WITH VOXELS

Constructing a volume representation from silhouette information is a pivotal step in deducing the 3-D shape of wear particles. 3-D reconstruction initiates by defining a bounding box $B$ as the maximal spatial domain. This can be achieved by selecting an appropriate voxel size. Voxels serve as the fundamental units in 3-D space, which are similar to pixels in 2-D images. Each voxel represents a small cubic volume, which can be employed to signify occupancy, i.e. 0 as 'unoccupied' and 1 as 'filled'. Each voxel is assigned a unique 3-D coordinate, collectively covering the entire space.

For every extracted silhouette of a wear particle, we project it from the image plane into 3-D space, which includes basic affine transformations, such as rotation and translation, to accurately reflect the spatial orientation of the wear particle. Each voxel is initially assigned a "filled" status, presuming that all voxels pertain to the wear particle. To eliminate voxels that do not correspond to the wear particle, SFS fully leverages the contours extracted from multiple binary silhouette images captured from different viewpoints. For each silhouette, SFS iterates over its 2-D pixels, marking the corresponding projected 3-D space as 'filled' or 'unoccupied'.

Specifically, any voxel that falls within the depth range of the silhouette is marked as 1, otherwise, it is marked as 0. This step is iteratively applied to all captured silhouette images, with each silhouette from a particular viewpoint filling corresponding voxels in the spatial region.

Note that the overlap among the voxel regions across multiple silhouettes typically utilizes a logical 'OR' operation. However, a logical 'OR' of extruded silhouettes would over-expand the volume. In contrast, in extremely harsh environments like jet engines, a strict logical AND would permanently carve away parts of the particle if a single view is flawed. Thus, we adopt a 'majority vote' strategy: a voxel is kept if it appears in at least half of the views. Otherwise, it is removed from the 3D shape. This hybrid strategy effectively prevents both the massive volume bloat of an OR and the catastrophic volume loss of an AND.

Upon completion of the scanning process using all silhouette images, the voxels marked as '1' form a 3-D reconstruction of the wear particle, primarily profiling its external shape. The level of detail in the reconstruction is significantly influenced by the size of the voxels. Smaller voxels provide higher resolution, allowing for a more precise representation of the wear particle. However, this increased resolution comes at the cost of high computational overhead. Similarly, the quality of the reconstruction relies on both the resolution of the silhouette images and the diversity of viewpoints utilized during capture. High-resolution silhouette images with clear boundaries, combined with a broad range of camera viewpoints, result in a more accurate 3-D reconstruction that effectively reflects the true shape of the wear particle.

Figure 9 presents the results of 3-D reconstruction of two wear particles from 2-D images under various camera viewpoints. The size of bounding box and voxel are configured to $50mm \times 50mm \times 50mm$ and $0.5mm \times 0.5mm \times 0.5mm$, respectively, which means each bonding box contains 1M voxels. We can clearly observe that as the number of applied 2-D images increases (i.e., with more camera viewpoints),

the level of details, such as edges and corners, in the 3-D representation of the wear particles enhances significantly, facilitating a more accurate reconstruction.

## 4. Implementation and Evaluation

### 4.1. Hardware Implementation

QuantWear integrates a power supply, optical components, and electro-optical system featuring distinct Na and K atomic vapor filters. Performance is optimized via specific thermal parameters ($220°C$ for Na, $125°C$ for K) and a $0.18T$ magnetic field. High-speed camera employs a Vision Research Phantom V211 (2200 fps, $1280 \times 800$, 32GB RAM), with lens parameters tuned to balance FoV and resolution, as detailed in Table 3 in Appendix C. The detailed hardware configuration of QuantWear is shown in Table 4 in Appendix C.

We conduct field tests on three solid fuel jet engines, as shown in Figure 10), with varying displacements (small, medium, large) and spout sizes during both day and night. The quantum optical system is deployed at a certain meters away from the engine, supported by an anti-vibration tripod. Core components of QuantWear, such as optical lens, atomic filter and high-speed camera, are fully encased by vibration-absorptive materials, i.e., sponge, to alleviate the impacts of vibration caused by intense jet flows.

### 4.2. Software Implementation

The SFS algorithm is implemented using Matlab, which using 18 projections ($20°$ angular resolution) to optimize the trade-off between details and computational overhead. Key parameters include a binarization threshold of 100 and a voxel resolution of $10^3$ pixels, whose threshold is set to 17 for profiling the 3-D wear particles. The dataset comprises 3.168 million images extracted from the central 8 minutes of 10-minute recordings. A modified YOLOv8 model is trained via PyTorch on an Intel i7-13700K/RTX 4090 server, utilizing Mosaic/MixUp augmentation and Dropout/Label Smoothing regularization.

In our work, we employ a strict temporally independent data split during the training of the modified YOLOv8 model to prevent temporal leakage. Specifically, for the same engine, an 8-minute (480 s) video is divided into the first 336 s and the last 144 s. For different engines, the split is performed based on separate physical test runs. In particular, the model is trained exclusively on data from one engine run, consisting of a continuous 7-minute sequence, and validated on a completely distinct and independent engine run, consisting of a continuous 3-minute sequence.

We use X-AnyLabeling, a state-of-the-art automatic labeling tool, to generate ground-truth bounding boxes for training

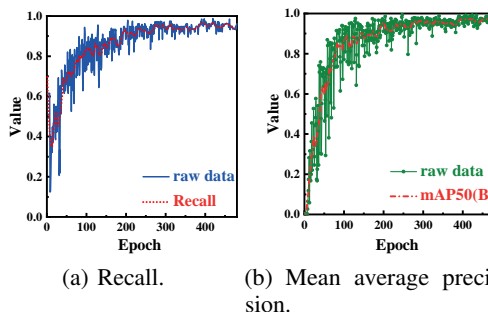

(a) Recall. (b) Mean average precision.

*Figure 12.* Wear particle enumeration using YOLOv8.

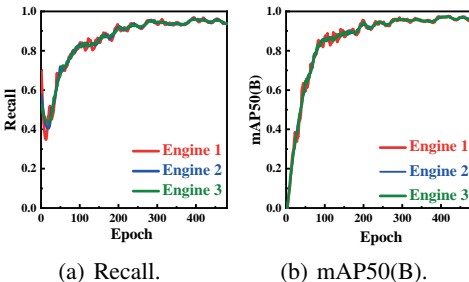

(a) Recall. (b) mAP50(B).

*Figure 13.* Performance on different engines.

our modified YOLOv8 model. This automated approach is highly reliable for our specific scenario because our optical system filters out background flame radiation, causing wear particles to appear as distinct bright spots against a dark background. More importantly, owing to the use of an ultra-high-speed camera (i.e., resolution of $1280 \times 800$ and pixel size of $25.6 \mu$m), consecutive frames exhibit a high degree of similarity, with particle motion being the primary variation between frames. Even sub-pixel-scale wear particles preserve sufficient spatial detail to be identified and labeled without ambiguity. Therefore, X-AnyLabeling produces highly accurate annotations as robust ground truth, requiring only minimal manual correction.

### 4.3. Evaluation Results

**Image Quality (SNR).** Our atomic filter significantly enhances image quality by suppressing flame background noise. The system achieves an average Signal-to-Noise Ratio (SNR) of **22.5dB**. In contrast, if we directly connect the optical lens to the high-speed camera to record the jet flows, as shown in Figure 11, the SNR of recorded images dramatically reduces to 2.9dB. The high SNR achieved by QuantWear validates the effectiveness of our quantum optical system, providing reliable 2-D images that can precisely capture the elemental Na and K.

**Wear Particle Enumeration.**

Detailed training curves and per-engine performance are shown in Figure 12 and Figure 13, respectively. The model achieves a Mean Average Precision (mAP) of **98.4%** and a Recall of **96.0%** after 500 training epochs. More im-

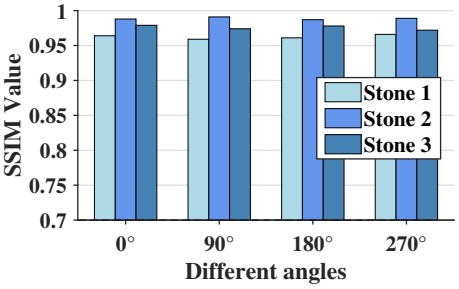

*Figure 14.* Performance on 3-D reconstruction.

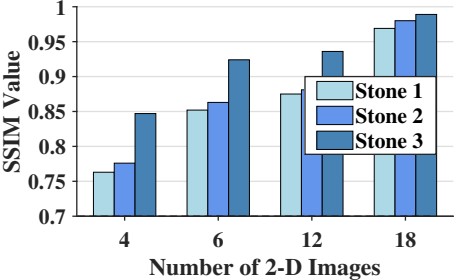

*Figure 15.* Performance on different number of images.

different viewpoints for each stone to perform 3-D reconstruction, each of which covers a view of 20 degrees. As shown in Figure 14, using the Structural Similarity (SSIM) (Huang et al., 2021) index on the simulated stones, the reconstructed 3-D shapes achieve an SSIM exceeding **95%** compared to the ground truth assets. The SSIM result indicates excellent performance of the 3-D reconstruction algorithm, which can accurately capture the shape of wear particles for jet engines from a full 3-D perspective.

**Performance on Number of 2-D Images.** The number of 2-D input images significantly impacts 3-D reconstruction quality. We measured the SSIM of three jet engine wear particles across four angular resolutions. As shown in Figure 15, as the number of 2-D images increases, SSIM improves because additional viewing angles provide richer silhouette data. Specifically, SSIM exceeds $95\%$ when using 18 images. Although higher resolution increases computation, we selected 18 images ($20°$ resolution) per particle. Since reconstruction occurs offline on a high-end server, computational delay is acceptable. Repeated tests with different 18-image batches yielded consistent SSIM values.

### 4.4. Performance on Different Backgrounds

In this evaluation, we separately record the jet flows from an identical engine under two distinct conditions: during the daytime with a bright background and at night with a dark background, all within the same test field. For each scenario, we capture a total of 500 images across three engines and measure the average SNR of images. The measured SNR for daytime and nighttime reaches 21.27dB and 23.68dB, respectively. This high image quality in both scenarios is due to our designed atomic filter, which mitigates undesired environmental light, accurately isolating Na and K spectra.

## 5. Limitations and Discussion

**System Generalization.** We notice that the constraint primarily lies in economic issues, not in methodology. High testing costs ($500K/test) currently limit our dataset to just three engines, restricting system generalization. To overcome this economic barrier, deep learning generalization can be improved by replacing expensive aerospace engines with affordable civilian ones, and by utilizing simulations like Unreal Engine 5 or NVIDIA Isaac Sim for complex fluid and particle dynamics.

**Baseline Evaluation.** We will compare practical baselines, including thresholding, blob detection on filtered Na/K images, and a non-atomic optical baseline, to clarify the contributions of atomic filtering, classical image processing, and modified YOLOv8.

**Deployment.** Future deployment will require optimized camera and nozzle placement, as engine size, nozzle geometry, and site layout can affect jet-flow visibility and particle imaging resolution. We will establish placement

portantly, the performance remains robust ($> 95\%$ mAP) across all three different engine types.

Table 2 compares 2D detection between standard YOLOv8 and the modified version with TR-C2f. The comparison result shows that our modified hybrid YOLOv8 outperforms standard YOLOv8. This is because TR-C2f integrates a Transformer Bottleneck Block with MHSA and a Feed-Forward Network to distinguish micro-particles from transient artifacts via global spatial correlation analysis. Placed at the backbone end, it preserves semantic features and improves detection stability under extreme conditions.

*Table 2.* Comparison between Standard YOLOv8 and Modified YOLOv8 (TR-C2f)

| Category | Metric | Standard YOLOv8 | Modified YOLOv8 (TR-C2f) |
|---|---|---|---|
| 2D Detection | mAP50 | 96.9% | 98.4% |
| | Recall | 94.2% | 96.0% |
| | Precision | 95.0% | 96.8% |

**3-D Reconstruction Accuracy.** Due to the inherent challenge of obtaining operational ground truth, we validate the 3D reconstruction via a high-fidelity hybrid simulation utilizing Unreal Engine 5 (UE5) and NVIDIA Isaac Sim. UE5 generates configurable jet engine environments, while Isaac Sim simulates particle trajectories under gravitational and fluid-dynamic forces. Randomly shaped stones, sourced from Sketchfab high-fidelity assets, serve as controllable physical surrogates. These assets are physically parameterized in Isaac Sim to emulate wear particle dynamics before importation into UE5 for photorealistic rendering. Our method's accuracy is validated by comparing algorithm outputs with perfect synthetic baselines.

During the simulation, 18 2-D snapshots are captured from

guidelines and calibration procedures to support stable and reproducible measurements.

**Ground Truth.** Future work will strengthen ground-truth validation using expert-verified Na/K annotations and controlled experiments with known particle properties. For 3-D reconstruction, we will further validate geometry with measurable physical particles and simulations.

**Small object detection.** Sub-millimeter high-speed particles remain challenging. Despite Na/K filtering, flame radiation, blur, and noise cause 1–2 pixel errors, affecting size estimation and SFS reconstruction. We will use tracking, uncertainty-aware boxes, and segmentation before fusion.

**Reconstruction Quality.** Concave or overlapping particles may underestimate volume and reduce counting accuracy, while low particle loads or weak Na/K signals may yield insufficient silhouettes for SFS. Future work will use multi-view imaging, Soft-NMS (density-aware suppression to improve 2-D pipeline's inference recall), confidence-aware silhouette selection, and temporal accumulation to improve robustness.

**Vibration Sensitivity.** Despite improved contrast from sC-MOS sensors and Na/K filters, engine vibrations may disrupt alignment and cause artifacts. Future work should assess vibration effects and refine calibration.

**Na/K Signal Dependence.** QuantWear's performance may degrade when Na/K signals are weak or inconsistent due to fuel chemistry or operating conditions. For engines using non-Na/K additives, keeping cross-engine robustness is difficult, as the atomic filter must be redesigned for corresponding elemental signatures.

## 6. Related Work

**Electrical Parameter-based method.** Electrical parameters such as current, inductance, and capacitance are sensitive to metal wear particles (Muthuvel et al., 2018; 2021; Xiaoliang et al., 2015; Jia et al., 2019; Kiaghadi et al., 2022; Hyka et al., 2021; Hwang et al., 2024; Um et al., 2024). For instance, Muthuvel et al. developed capacitive sensors integrated with permanent magnets to detect ferrous fragments via effective area alteration (Muthuvel et al., 2018), and later proposed passive wireless LC sensors to monitor resonance frequency variations (Muthuvel et al., 2021). To measure particle size, inductive coil arrays have been utilized as independent sensing channels (Xiaoliang et al., 2015). Furthermore, Jia et al. combined inductive pulse sensors with the Empirical Mode Decomposition with Reverse Reconstruction (EMD-RRC) algorithm to enhance detection accuracy and SNR (Jia et al., 2019). However, these technologies primarily detect the presence of metal fragments on static inner surfaces in normal environments, failing to enumerate or profile the 3-D shape of wear particles in high-speed jet flows.

**Vision-based method.** Deep learning and computer vision have been increasingly applied to particle tracking (Wei et al., 2024; Lee & Choi, 2024; Wei et al., 2023; Yun et al., 2024; Guan et al., 2022; Zhao et al., 2024; Jiang et al., 2021; Ren et al., 2023; Xu et al., 2024; Yuan et al., 2023; Cao et al., 2024). While deep learning has revolutionized industrial inspection, detecting minute particles remains challenging. Recent advances in small object detection (Wei et al., 2024) and industrial anomaly localization (Lee & Choi, 2024) have achieved high precision in static environments. However, these vision-based methods typically require stationary subjects and ontrolled lighting (Wei et al., 2023). They suffer from motion blur and ambient light interference for high-speed objects in bright combustion environments, making them unsuitable for jet flow monitoring.

**Optical-based method.** Optical sensors offer high resolution and precision (Hong et al., 2015; Gallego et al., 2020; Xie et al., 2024; Li et al., 2024a; Lee & Gilbert, 2025; Xian et al., 2024; Fan et al., 2023; Matsuo & Yamakawa, 2023; Cao et al., 2023; Zhang & Yu, 2022; Chung et al., 2022b). Innovations include debris sensors incorporating dual light sources (Hong et al., 2015) and entangled photon quantum imaging systems for weak-turbulence environments (Cao et al., 2023). Event-based sensors (Gallego et al., 2020) excel in high-speed tracking with microsecond latency, while computational imaging (Xie et al., 2024) targets reconstruction through scattering media. However, traditional optical methods lack the spectral selectivity to completely isolate wear particles from the intense, broad-spectrum radiation of jet flames. QuantWear addresses these limitations by using atomic-level quantum sensing to isolate wear-particle signals from complex environmental noise.

## 7. Conclusion

This study presents a holistic design and implementation of the first wear particle monitoring system for jet engines based on quantum optical technology. To accurately monitor the rapidly moving and rotating tiny wear particles in high-speed jet flows and bright environments, this study comprehensively designs a quantum optical system that innovatively enables the tracking of wear particles by imaging the elemental Na and K in the jet flows. The designed atomic filter effectively isolates the light generated by jet flows that align with a spectrum of Na and K atoms, while significantly alleviating light at undesired wavelengths. QuantWear enables enumeration and 2-D shape capturing of wear particles by fully leveraging their electrochemical reaction with elemental Na and K in high-temperature environments and employing YOLOv8 to accurately count their number and acquire their 2-D shapes. Finally, 3-D shapes of wear particles are reconstructed from 2-D images using the extracted silhouette information. The experimental results demonstrate the effectiveness of our QuantWear system, which offers reliable information for jet engine health monitoring.

## Acknowledgments

This work is supported by the Science and Technology Innovation Program of Hunan Province under Grant 2024RC3105 and 2025RC1031, the Guangdong Basic and Applied Basic Research Foundation under Grant 2024A1515011687, the Shenzhen Science and Technology Program under Grant JCYJ20240813162405008, the Financial Support of Lingnan University (SDS24A17), and the Hong Kong GRF under Grant 15206123. The corresponding authors are Yanwen Wang and Di Wu.

## Impact Statement

This paper presents work whose goal is to enable early diagnosis of jet engine damage by accurately detecting, counting, and reconstructing the 3-D shapes of wear particles in high-speed jet flows. There are many potential societal consequences of our work, most of which are related to the safety, reliability, and maintenance of jet-engine-powered systems, such as commercial aviation, manned spaceflight and turbines.

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

## A. Theoretical Background

### A.1. Wear Particles

Wear particles are microscopic metal fragments generated during the operation of mechanical systems due to friction and wear among vibrating components. Specifically, components such as bearings, pistons, and turbines within jet engines experience significant stress and heat in high-temperature environments (Murugan et al., 2017). As a result, these components gradually wear down over time, releasing tiny particles typically composed of metal and alloys, with sizes ranging from micron to millimeter scale. Wear particles are often expelled from the engines' spouts at extremely high speeds, propelled by the high-pressure and high-velocity jet flows. These wear particles move and rotate rapidly within the jet flows, exhibiting characteristics of extremely tiny sizes and high-speed motion, rendering them challenging to capture in the harsh working conditions of jet engines (Xu et al., 2017).

The characteristics of wear particles are essential for assessing engine health. By analyzing the size, shape, composition, and quantity of these particles, it is possible to understand the wear conditions of the engine and identify potential failures, thus preventing casualties and property loss from engine explosion (Mishra et al., 2017). For instance, certain types of wear particles may indicate abnormal abrasion of specific components, while an increasing quantity of wear particles may indicate that a component is approaching failure. Therefore, capturing and analyzing these wear particles not only aids in determining the current operational status but also provides critical indicators for future failure diagnosis.

### A.2. Quantum Jump Mechanism

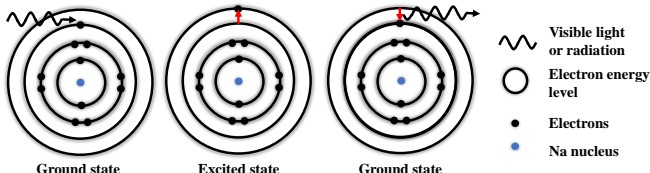

*Figure 16.* Quantum jump for sodium (Na) atoms.

Quantum jump, also known as atomic electron transition, is a fundamental concept in quantum mechanics, describing the process in which an electron jumps from a lower energy level to a higher one, as shown in Figure 16. Given Na atom as an example, an electron must absorb a photon whose energy $E_{\text{photon}}$ is equal to the energy difference $\Delta E$ between the two levels for transition (Gallo et al., 2016): $E_{\text{photon}} = hv = E_2 - E_1$, where $E_{\text{photon}}$ is the energy of the photon, $h$ is Planck's constant, $v$ is the frequency of the photon, and $E_2$ and $E_1$ are the energies of the higher and lower energy levels, respectively. Conversely, when an electron transitions from a higher energy level to a lower one, it emits a photon with the same energy. Quantum jump builds a foundation for many physical phenomena, such as the working principles of quantum sensing (Gajland et al., 2024; Jankiewicz et al., 2022; Germanakos, 2024; Ashktorab et al., 2019), quantum computing (Bayerstadler et al., 2021; Kim, 2024; Li et al., 2024b; Chung et al., 2022a), quantum communication (Kasi & Jamieson, 2020), quantum cryptography (Bozzio et al., 2022; Bennett et al., 1992), and spectroscopic analysis (Pelton & McLean, 2000; Forseth & Schroeder, 2011), playing a crucial role in understanding of the interaction between light and matter (Chen et al., 2021). In our study, we employ quantum jump to absorb and emit photons, selecting light in specific spectrum to capture the movement of elemental Na and K in the jet flow.

### A.3. Faraday Rotation

Faraday rotation is a non-reciprocal physical magneto-optical phenomenon that describes the polarization rotation of linearly polarized light as it traverses a medium subjected to a magnetic field (Crassee et al., 2011). When source light traverses a magnetic field, the polarization angle $\theta$ is directly proportional to the projection of the magnetic field $\vec{B}$ along the direction of the light propagation, which can be calculated as $\theta = VBd$, where $V$ is the Verdet constant, $B$ denotes the magnetic field strength, and $d$ is the propagation length within the medium. The Verdet constant $V$ varies with wavelength, material, and temperature. Given a specific medium with a certain length, the rotation of polarization angle for light waves at typical wavelengths can be precisely controlled such that the desired light in jet flows can be effectively extracted. We place two customized polarizers before and after the atomic filter, allowing the light waves of Na and K with corresponding polarization to pass through while blocking light with other polarization.

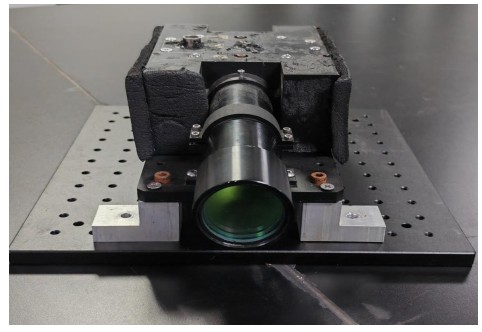

(a) Lens (front view).

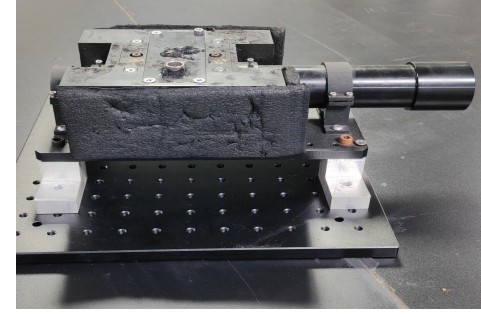

(b) Lens (side view).

*Figure 17.* Optical lens details.

## B. Validation of Quantum Optical System

We employ a high-end fiber optic spectrometer (FOS) to validate the effectiveness of our designed quantum optical system. The FOS is connected to the output of the atomic detector for real-time spectrum analysis, which can be applied to validate the desired spectrum of light, enabling successful imaging for desired atoms. During experiment, we ignite the jet engine for 10 minutes and use our quantum optical system to capture the jet flow, recording the light emission characteristics typically in the spectrum of Na and K. During the experiment, we vary the temperature of vapor cell from low to high while maintaining the magnetic field intensity.

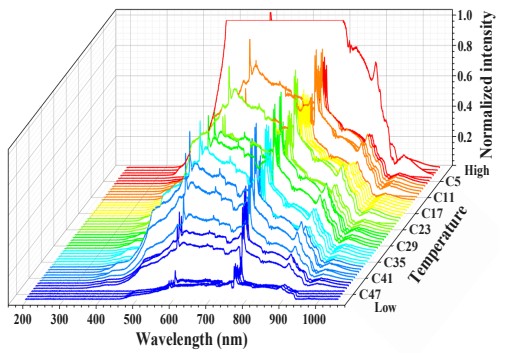

(a) Spectral diagram of Na and K atoms.

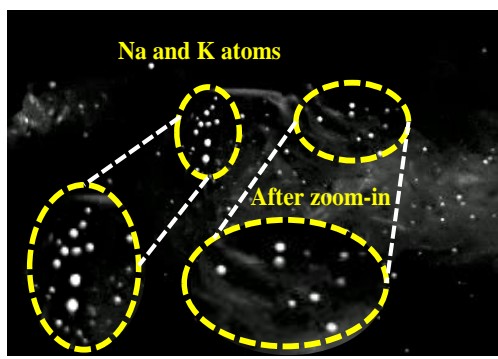

(b) The captured elemental Na and K.

*Figure 18.* Validation of our quantum optical system: (a) Spectral response at different temperatures; (b) A sample captured frame showing Na and K distribution.

Figure 18(a) depicts the measured spectrum of the light after passing through the atomic detector at 50 temperatures. Most of the output spectrum exhibits two distinct peaks corresponding to specific wavelengths of 589nm for Na atoms and 766nm for K atoms, respectively. When we zoom in the two peaks, the spectrum reveals a clear bi-spectral peak pattern. Both the wavelengths and bi-spectral patterns exactly match the spectral characteristics of Na and K atoms. In addition, light at other wavelengths is successfully attenuated, resulting in a relatively lower energy spectral pattern. We observe that a lower temperature corresponds to a decreased energy of photons in the Na and K spectra since the number of active atoms in the vapor cell is insufficient to effectively interact with the light, leading to a reduced total energy of re-emitted photons after quantum jumps. In contrast, excessively high temperatures incur a broader transmission band, allowing a wide range of wavelengths to pass through the atomic filter. As the temperature increases, intense thermal motion of atoms leads to an increase in Doppler broadening (Firstenberg et al., 2008). As such, the transmission peaks near the resonance wavelength become more diffuse and no longer concentrated at a sharp peak, forming a band-shaped pattern.

Figure 18(b) illustrates a single frame of image captured from the jet flow when the vapor cell is configured at a proper temperature and magnetic field intensity. Multiple light spots are randomly distributed throughout the jet flow, accurately representing the presence of elemental Na and K. In addition, the dim vaporous pattern in the background indicates other wavelength of light that are not entirely annihilated by our atomic filter. By unceasingly acquiring images of jet flow using a high-speed camera, we can effectively capture the movement of elemental Na and K in the jet flow.

## C. Hardware Configuration of QuantWear

*Table 3.* Technical Parameters of Lens Design

| Item | Design | Measurement |
|------|--------|-------------|
| System Focal Length | 50 mm | 50 mm |
| F# | 5 | 5 |
| Working Wavelength | 589±0.1 nm | 589 nm |
| Azimuthal Field Angle | 10.5° | 10.47° |
| Inclination Field Angle | 10.5° | 10.47° |
| Diagonal Field Angle | 14.9° | 14.77° |

Table 3 summarizes the key technical parameters of the designed imaging lens and the corresponding measured results. The experimental measurements are consistent with the design specifications. The system achieves a focal length of 50 mm with an F-number of 5 at the operating wavelength of 589 nm. In addition, the measured azimuthal, inclination, and diagonal field angles are all close to the theoretical values, indicating that the optical system satisfies the intended imaging performance requirements.

*Table 4.* Atomic Filter System Components

| Component | Function | Key Parameters |
|-----------|----------|----------------|
| Power | Voltage/current | 0–15 V, 0–5 A
Temp ±0.1 °C |
| Atomic Filter | Na FADOF | 589 nm, 0.01 nm BW
70–80% trans, 200–300 G
80–280 °C |
| Optical Lens | Imaging | 50–150 mm focal, f/2.8–f/5.6 |
| Filter Controller | Controller | 0–300 °C, ±0.1 °C
Real-time display |

Table 4 lists the main components of the atomic filter imaging system and their corresponding operating parameters. The system consists of a power supply, Na-FADOF atomic filter, imaging lens, and filter controller. Key parameters such as operating voltage, temperature stability, magnetic field strength, transmission bandwidth, and focal length are provided, demonstrating the integrated configuration and operating conditions of the experimental platform.

