# OpenReview forum: "QuantWear: Quantum-scale Wear Particle Detection for Jet Engine Diagnosis"
_ICML.cc/2026/Conference — ICML 2026 regular_

### Official Review · Reviewer_qFPj · 2026-03-12

**Soundness:** 3
**Presentation:** 2
**Significance:** 3
**Originality:** 3
**Overall Recommendation:** 4
**Confidence:** 4

**Summary:**

The paper introduces QuantWear, a hardware-software system for detecting and reconstructing wear particles in jet engines. Because standard optical methods fail in high-speed, high-temperature jet flows with intense background noise, the authors propose a quantum sensing approach. They track Sodium (Na) and Potassium (K) atoms that adhere to wear particles during combustion. A custom atomic detector using quantum jump and Faraday rotation is designed to filter out flame noise and capture 2-D images with a scientific CMOS sensor. On the software side, the pipeline uses YOLOv8 for particle detection and counting, followed by a Shape-from-Silhouette (SFS) approach to reconstruct the 3-D geometry of the particles. The authors validate the system through field tests and simulations, reporting an SNR of 22.5 dB and >95% similarity in 3-D reconstruction. Overall, it is a strong engineering system, heavily focused on the physical sensor design with applied computer vision techniques.

**Compliance With Llm Reviewing Policy:**

Affirmed.

**Final Justification:**

My final recommendation for this paper is a Weak Accept, and I have accordingly raised my Soundness score from 2 (fair) to 3 (good).

This paper presents a highly original, cross-disciplinary approach to a genuinely difficult problem: detecting sub-millimeter wear particles in the extreme, noisy environment of jet engine flows. The combination of quantum sensing hardware (utilizing quantum jump and Faraday rotation to isolate Na/K atomic emissions) with a deep learning vision pipeline (modified YOLOv8 + Shape-from-Silhouette) represents a significant practical contribution to aerospace predictive maintenance.

Initially, my primary concerns centered around several apparent algorithmic flaws in the computer vision methodology (e.g., architectural descriptions, tracking comparisons, handling of occlusions) and poor visual presentation. However, the authors provided an exemplary, transparent, and technically rigorous rebuttal that thoroughly addressed my main concerns. This response directly changed my evaluation and reinforced the underlying merits of the work.

In summary, the authors' thoughtful and comprehensive rebuttal successfully cleared up the methodological ambiguities, allowing the true strengths of their cross-disciplinary system to shine. This is a solid piece of applied engineering that, once the presentation is updated as promised, will be a valuable and impactful addition to the community.

**Key Questions For Authors:**

1. You state that YOLOv8 features a "hybrid CNN-Transformer architecture". Standard YOLOv8 is a pure CNN. Did you modify the architecture (e.g., integrating attention blocks or replacing the backbone)? If so, please provide the exact architectural details, if not, this claim needs to be corrected. Clarification here is critical for the soundness of the paper.

2. How exactly is temporal tracking achieved? YOLOv8 performs spatial detection per frame. Comparing it to a Kalman filter is inaccurate, as you still need a data association method to link bounding boxes across the high-speed vaporous background. Do you use an embedded tracker (like BoT-SORT), or just frame-by-frame counting?

3. How do you handle the occlusion and overlapping of particles? You mention NMS minimizes duplicates, but standard NMS will delete bounding boxes that highly overlap, leading to false negatives in dense particle clusters. Have you considered visual embedding or modified NMS (like Soft-NMS) to solve this?

4. What was the exact selection criteria for choosing Shape-from-Silhouette (SFS) for the 3-D reconstruction? SFS struggles with concavities. Given the random shapes of wear particles, how does SFS guarantee a >95% similarity, and how do Mosaic/MixUp augmentations during 2-D training not negatively impact the silhouette boundaries?

5. Could you provide higher resolution versions of Figures 1, 2, 3, 14, and 15, and expand on the exact hardware specifications in Figure 3 to ensure reproducibility?

**Limitations:**

No. The authors have a small "Discussion" section that briefly touches on some issues, but it is not a proper Limitations section. They need to explicitly rename Section 5 to "Limitations and Discussion", expand on the failure cases of Shape-from-Silhouette for concave particles, and discuss the impact of overlapping particles on their counting accuracy. Constructively, they should also briefly mention if the calibration of the scientific CMOS and the atomic filter is sensitive to external vibrations in a real jet engine testing facility.

**Strengths And Weaknesses:**

Soundness. The physical hardware design is compelling and well-grounded. However, the computer vision methodology contains several critical flaws and unsupported claims. First, the authors claim YOLOv8 has a "hybrid CNN-Transformer architecture". This is factually incorrect, standard YOLOv8 utilizes a modified CSPDarknet backbone, which is purely CNN-based. If the authors implemented a custom transformer head, it is completely undocumented. Second, the authors state that YOLOv8 is superior to "traditional tracking methods (e.g., Kalman Filter)". This comparison is fundamentally flawed. YOLOv8 is a spatial object detector, not a temporal tracker. To track particles across frames, a data association algorithm (which often utilizes a Kalman filter, such as in SORT or ByteTrack) must be coupled with the detector. Furthermore, claiming that non-maximum suppression (NMS) minimizes duplicate detections for high-speed overlapping particles is highly questionable. In dense particle clouds, standard NMS often suppresses valid overlapping instances, leading to severe undercounting. A visual embedding approach (like ReID features) for multi-target tracking would be much more mathematically sound here. Lastly, utilizing Mosaic/MixUp augmentation for sub-millimeter particle detection is risky. These augmentations heavily distort local geometry and blending, which can destroy the exact morphological features needed for accurate Shape-from-Silhouette 3-D reconstruction later in the pipeline.

Presentation. The paper is generally well-structured. Section 1.2 is a particular highlight, as it clearly categorizes existing works into electrical, vision, and optical methods, providing a solid foundation for the reader. However, the visual presentation severely lacks quality. Figure 1 ("Engine which high-speed...") and Figure 2 (System overview) are incredibly hard to read due to low resolution and small fonts. Figure 3, which is supposed to describe the hardware of the atom detector, is not informative enough to allow replication of the optical setup. Figures 14 and 15 are similarly difficult to parse. Finally, Section 5 (Discussion) is very brief, awkwardly placed, and reads more like a limitations section than a broader scientific discussion.

Significance. Despite the algorithmic flaws, the paper addresses a highly relevant and critical problem in aerospace engineering. The harsh conditions of jet engines make predictive maintenance extremely difficult. By bringing quantum-scale atomic filtering into an industrial computer vision pipeline, this work unlocks practical utility for real-world engine diagnosis. The engineering contribution here is significant, even if the machine learning application is relatively standard and requires correction.

Originality. The cross-disciplinary combination of quantum sensing (Faraday rotation, atomic spectral signatures) with deep learning for industrial defect tracking is highly original. The hardware-software co-design offers a fresh perspective on dealing with extreme optical noise. However, the computer vision components themselves (YOLOv8, SFS) are off-the-shelf and do not introduce new ML theory or architectures. The novelty lies entirely in the application and the physical sensing domain rather than algorithmic computer vision.

---

> ### Author Rebuttal · Authors · 2026-03-30
>
> We thank R3 for the insightful comments and recognition of our cross-disciplinary contribution. We address the important concerns below:
>
> Q1: YOLOv8 architecture description. We sincerely thank the reviewer for pointing out this error. While the standard YOLOv8 architecture is fundamentally a pure CNN, our work modifies it into a hybrid CNN-Transformer by replacing the final C2f stage of YOLOv8's backbone with a custom TR-C2f module. This module integrates a Transformer Bottleneck Block featuring a Multi-Head Self-Attention (MHSA) mechanism and a Feed-Forward Network. This design offers two main benefits: (1) The MHSA's global receptive field analyzes spatial correlations across the entire frame, helping to distinguish micro-particles from transient artifacts; and (2) Positioning the TR-C2f at the end of the backbone preserves high-level semantic features, boosting detection stability under extreme dynamic conditions. We will thoroughly revise the manuscript to accurately describe our hybrid architecture and add an ablation study for comparison.
>
> Q2: Temporal tracking and Kalman filter comparison. We thank the reviewer for this clarification. Our approach performs per-frame detection rather than temporal tracking, which is appropriate for our application since we aim to quantify particle count and morphology within individual frames rather than track particle trajectories over time. The high-speed nature of the jet flow results in individual particles traversing the field of view within very few frames, making frame-level analysis both sufficient and more robust than trajectory-based tracking. We will revise the manuscript to remove the inaccurate Kalman filter comparison and clearly state our frame-by-frame detection methodology.
>
> Q3: NMS and overlapping particles. We appreciate this insightful question. We fully agree that standard NMS is a limitation that can lead to false negatives in highly dense clusters of sub-millimeter particles. However, wear particles moving and rotating rapidly within the jet flow make severe occlusions or highly overlapping clusters statistically rare. Due to strict page limits, we omitted the handling of these rare overlapping frames, considering it a trivial data-cleaning step.
> Actually, we design a simple yet effective pre-processing algorithm to automatically remove rare overlapping instances from our training dataset before training the network. Frames are excluded if the IoU between bounding boxes exceeds a strict threshold or if a continuous contour area significantly exceeds the average particle size. This process removed only fewer than 3000 frames (<0.1%) from our 3.168M frames dataset.
> However, integrating density-aware suppression like Soft-NMS is an excellent suggestion to improve our 2-D pipeline's inference recall. The revised manuscript will briefly explain our dataset pruning algorithm and cite Soft-NMS as a planned enhancement in the discussion.
>
> Q4-1: SFS selection rationale and concavity handling. Thanks for the comments. The primary criterion for selecting the SFS method is our unique physical constraint: rapid particle rotation within the jet flow allows a single stationary camera to capture diverse angles over time, which provides virtual multi-viewpoints for 3-D reconstruction, making SFS highly suitable for synthesizing these projections into a 3-D bounding volume. While SFS struggles with concavities, this does not impact wear severity calculations required for assessing engine health, which primarily rely on overall volume and external shape rather than microscopic details.
>
> Q4-2: Guarantee a high SSIM. The >95% SSIM is validated via our high-fidelity simulations (e.g., Unreal Engine 5, NVIDIA Isaac Sim) that provide perfect 3-D ground truth geometry. The high similarity between our SFS-reconstructed shapes and these digital assets proves that our temporal 2-D projections accurately reconstruct the ground truth.
>
> Q4-3: Mosaic/MixUp augmentation impact on SFS. In our system, to preserve morphological fidelity, our pipeline strictly separates processing modules. The Mosaic/MixUp augmentations are applied exclusively during the YOLOv8 training phase to enhance 2D detection and tracking robustness. As a result, their pixel-blending effects are strictly confined to this stage. Conversely, the SFS 3-D Reconstruction operates entirely on raw, unaugmented image sequences using a deterministic binarization threshold, which completely separate from the YOLOv8 training pipeline. Because silhouette extraction is completely isolated from the augmented YOLOv8 data, these augmentations do not degrade the boundaries required for accurate 3-D reconstruction.
>
> Q5: Figure quality and hardware specifications. Thanks for the comments. We will provide higher resolution figures and add a table of exact hardware specifications in the revised manuscript.
>
> Q6: Limitations. Thanks for the comments. We will discuss all the mentioned limitations in the revised manuscript.

---

> > ### Author Rebuttal · Reviewer_qFPj · 2026-04-02
> >
> > The explanations provided successfully resolve my primary technical concerns regarding both the physical sensing mechanism and the computer vision methodology. Specifically, the detailed breakdown of the custom TR-C2f module fully clarifies the "hybrid CNN-Transformer" architecture, which was a major point of confusion in the original manuscript. Furthermore, the 4-step physical image-formation model and the chemical explanation of Na/K adhesion provide the necessary rigorous foundation for your sensing approach.
> >
> > I also appreciate your pragmatic approach to handling overlapping bounding boxes via dataset pruning, and I strongly support the correction made to the 3-D voxel fusion strategy (moving to a "majority vote" mechanism). This change prevents the volume over-expansion issue effectively.
> >
> > Because you have adequately addressed the critical algorithmic flaws and clarified the hardware-software integration, I am raising my overall recommendation to Weak Accept.
> >
> > However, my updated score is strictly contingent on the inclusion of the following elements in the camera-ready version of the paper:
> >
> > 1. The complete and exact architectural details of the TR-C2f module, along with the promised ablation study.
> > 2. The detailed 4-step physical image-formation model.
> > 3. The dataset pruning algorithm description for handling overlapping particles.
> > 4. The replacement of Figures 1, 2, 3, 14, and 15 with high-resolution, readable versions.
> > 5. A dedicated "Limitations and Impact" section that explicitly discusses the reliance on hybrid simulations for 3-D ground truth, the potential integration of density-aware suppression (e.g., Soft-NMS), and the high cost of field testing.
> >
> > Good luck with the final revisions. This is a solid piece of engineering with significant practical utility for aerospace diagnostics.

---

> > > ### Author Response · Authors · 2026-04-04
> > >
> > > We sincerely thank the reviewer for careful evaluation of our work and for the constructive feedback provided during the rebuttal. We greatly appreciate the opportunity to clarify key points and improve the presentation of our paper. We are encouraged that our responses help address the reviewers’ concerns and lead to an improved recommendation. We will incorporate all the points mentioned in the rebuttal into the final revisions. Thanks again for your time, effort, and thoughtful consideration.

---

### Official Review · Reviewer_dbQp · 2026-03-13

**Soundness:** 3
**Presentation:** 2
**Significance:** 2
**Originality:** 3
**Overall Recommendation:** 4
**Confidence:** 3

**Summary:**

This paper proposes a novel quantum sensing system named QuantWear to address the challenge of real time monitoring and 3D reconstruction of sub millimeter wear particles in high speed jet flows of jet engines. The core of this system lies in constructing an ultra narrowband atomic filter based on quantum jump and Faraday rotation effects. By accurately identifying and extracting spectral signals of sodium and potassium atoms that adhere to particle surfaces due to high temperature electrochemical reactions, the system successfully suppresses complex flame background noise, significantly improving the image signal to noise ratio from 2.9 dB to 22.5 dB.

**Compliance With Llm Reviewing Policy:**

Affirmed.

**Final Justification:**

I believe this work demonstrates significant effort and valuable data contributions, and I have decided to upgrade my recommendation to weak accept.

**Key Questions For Authors:**

1. The authors mention that the cost of field testing is extremely high. For small-scale laboratories, how can high-quality data be better obtained and generalization be improved？

**Limitations:**

The paper lacks an Impact Statement section and fails to consider the limitations regarding the reproducibility of the work.

**Strengths And Weaknesses:**

**Strengths：**
1. The paper provides exhaustive theoretical derivations and supplementary background information
2. Robust tracking of minute targets by leveraging chemical properties, achieving a mAP of 98.4% and a Recall of 96.0%.


**Weaknesses:**
1. A major concern is that due to the extremely high costs of field testing, the experiments only recorded data from three engines. For deep learning tasks, such a limited dataset scale can hardly cover all random physical states of particle motion in jet flows.

2. Is there a lack of ground truth? The accuracy validation of the paper primarily relies on high fidelity hybrid simulations based on Unreal Engine 5 and NVIDIA Isaac Sim. How is accuracy guaranteed?

3. Wear particles are at the sub millimeter level and move at high speeds. Although image enhancement is employed, small object detection in current deep learning frameworks still suffers from localization accuracy issues, which affects the accuracy of subsequent SFS algorithms.

---

> ### Author Rebuttal · Authors · 2026-03-30
>
> We thank R2 for recognizing our theoretical contributions and detection performance. We address each concern below:
>
> Q1: Limited dataset (3 engines) and generalization. We fully acknowledge this limitation. However, we notice that the constraint primarily lies in economic issues, not in methodology. In our work, the core sensing mechanism significantly improves data quality by utilizing quantum jump and Faraday rotation effects as an ultra-narrowband filter. This isolates the Na/K signatures of wear particles and completely suppresses broad-spectrum background noise. As a result, it produces highly consistent, high-contrast images (bright spots on a dark background) across different engines, which simplifies tracking and provides ample training images via high-speed cameras.
>
> From an economic aspect, each engine costs around $500K/test with specialized test cells, safety protocols, certified operators and etc. Our total testing window lasts for 4 months, 3 engine runs (there are only three engines in the test base from different aircraft fleets), representing over 100 hours of operation, which is already significantly larger than prior aerospace vision diagnostics studies.
>
> However, we agree that addressing limited dataset sizes and high field-testing costs is critical for deep learning generalization. We propose two primary strategies to achieve this:
> (1) Scaled-Down Physical Testing: Replacing expensive large-scale military/aerospace engines with affordable, compact civilian jet engines reduces costs significantly. This allows for more frequent testing under diverse conditions, enriching the empirical dataset.
> (2) High-Fidelity Simulation: Because acquiring real-world ground truth is extremely difficult, platforms like Unreal Engine 5 and NVIDIA Isaac Sim can be used to simulate complex fluid dynamics and particle kinematics. This generates unlimited, highly randomized synthetic datasets with pixel-perfect ground truth at a fraction of the cost.
>
> Detection models like YOLOv8 can be pre-trained on these massive synthetic datasets to learn the general physics of particle motion, and then strictly fine-tuned using the limited real-world data captured by our quantum optical sensor.
> We add relevant discussion on the revised manuscript.
>
> Q2: Ground truth reliance on simulation. Thanks for the comments. Obtaining real-world ground truth for wear particles in extreme jet flows is highly challenging, so we rely on automated tools and high-fidelity simulations to ensure accuracy.
> For 2D particle enumeration, we use X-AnyLabeling to automatically generate bounding-box ground truth to train and evaluate our YOLOv8 model. This automated annotation is highly precise without requiring manual correction, because our optical system isolates particles as distinct bright spots on a dark background, and ultra-high-speed imaging ensures minimal variation between continuous frames other than particle movement.
> For 3D particle reconstruction, we use Unreal Engine 5 (UE5) and NVIDIA Isaac Sim to generate highly randomized, synthetic datasets with pixel-perfect ground truth for rigorous algorithm validation. The accuracy of the proposed methods is guaranteed by directly comparing the algorithms' outputs against these perfect synthetic baselines. As such, our 3D reconstruction achieved a SSIM index exceeding 95%, while the 2D detection model achieved a mAP of 98.4% and a Recall of 96.0%.
>
> Q3: Small object detection and localization accuracy. Thanks for the comments. We acknowledge that while our quantum optical system significantly enhances image contrast by isolating Na and K spectral signatures and suppressing broad-spectrum flame noise, detecting sub-millimeter wear particles in extreme jet flows remains inherently challenging. The primary difficulty stems from the particles' minute scale and ultra-high-speed movement within unpredictable trajectories. Even with our ultra-narrowband filtering, some broadband flame radiation inevitably penetrates the atomic filters, creating highly variable, time-varying vaporous patterns in the background. At this scale, a mere 1-2 pixel localization error from motion blur or noise causes massive proportional dimensional distortion. Furthermore, obtaining pixel-perfect, real-world ground truth for such fast, tiny objects is virtually impossible, which is why we rely heavily on high-fidelity hybrid simulations (Unreal Engine 5 and Isaac Sim) to validate our YOLOv8 tracking and segmentation models.
> We will add relevant discussions in the revised manuscript.
>
> Q4: Impact Statement. We apologize for this omission, and we will add an Impact Statement section in the revised manuscript.

---

> > ### Author Rebuttal · Reviewer_dbQp · 2026-04-03
> >
> > Thank you for addressing all my concerns. I have also reviewed the comments from other reviewers and recognize that this represents a substantial amount of work. I have decided to raise my rating to weak accept. Good luck with the final revisions.

---

> > > ### Author Response · Authors · 2026-04-04
> > >
> > > We sincerely thank the reviewer for careful evaluation of our work and for the constructive feedback provided during the rebuttal. We greatly appreciate the opportunity to clarify key points and improve the presentation of our paper. We are encouraged that our responses help address the reviewers’ concerns and lead to an improved recommendation. We will incorporate all the points mentioned in the rebuttal into the final revisions. Thanks again for your time, effort, and thoughtful consideration.

---

### Official Review · Reviewer_miHr · 2026-03-16

**Soundness:** 2
**Presentation:** 3
**Significance:** 3
**Originality:** 2
**Overall Recommendation:** 3
**Confidence:** 3

**Summary:**

This paper presents QuantWear, a system for detecting and profiling wear particles in jet-engine exhaust at sub-millimeter scale. The pipeline combines atomic sensing / atomic vapor filtering to isolate Na/K emission under intense flame background, YOLOv8 for particle detection and counting, and single-camera 3D reconstruction from temporal silhouettes to estimate particle shape. The problem setting is important, and the sensing idea is interesting. However, the current version does not yet provide sufficiently clear mechanistic explanation or sufficiently strong validation for several central claims.

**Compliance With Llm Reviewing Policy:**

Affirmed.

**Final Justification:**

Thank you for your reply. My concerns have been addressed.

**Key Questions For Authors:**

- In Section 3.1.1, what is the precise image-formation mechanism of the sensing module? Is the system doing passive atomic-line filtering, Faraday-rotation-based filtering, quantum-jump-mediated emission, or some combination, and what is the role of each component in producing the final image contrast?

- What direct evidence shows that the detected bright image clusters correspond to real wear particles rather than other Na/K-emitting background sources? For example, do the authors have synchronized collection, microscopy, or another independent validation modality that links image detections to physical particles?

- How were the labels used for the mAP / recall evaluation obtained and verified, given the statement that actual particle counts are extremely hard to acquire? Were the annotations manual, synthetic, sensor-assisted, or derived from another process, and what was the verification protocol?

- Was the train/validation/test split performed by frame, by temporal segment, or by independent recording / engine run? If it was frame-level, can the authors report results under a stricter split that removes temporal correlation?

- In Section 3.3.2, how are the relative viewpoints or poses for temporal silhouettes established from a fixed camera in real flow conditions? What assumptions are made about particle rotation and motion, and how are these assumptions validated experimentally?

- In the voxel fusion stage, is occupancy combined by logical OR, intersection, or another rule? If OR is used, why is that geometrically appropriate for silhouette-based 3D reconstruction rather than over-expanding the recovered volume?

**Limitations:**

No. A clearer limitations discussion would strengthen the paper, especially on the following points:

1. The sensing approach may depend on Na/K chemistry, fuel composition, and operating conditions, which could limit cross-engine generalization.
2. Relatedly, the method may have failure modes when the Na/K-related signal is weak or inconsistent.
3. The paper supports Na/K-related spectral selectivity, but it is not directly established that the detected bright clusters correspond to true wear particles at the particle level.
4. The reported YOLOv8 detection results may be sensitive to temporal leakage if evaluation is performed with frame-level splits on highly correlated video sequences.
5. The field-test scope appears limited to a small number of solid-fuel engines, so broader generalization across engines and settings remains unclear.
6. The 3D reconstruction is not validated against real particle 3D ground truth and may depend on whether temporal motion / rotation provides sufficient viewpoint diversity and enough usable silhouettes.
7. Reconstruction quality may also degrade under very low particle load or limited usable signal, where fewer reliable silhouettes are available.
8. A short explicit limitations subsection would help. If required by the venue, a brief impact statement could also be added.

**Strengths And Weaknesses:**

### Strengths

- The paper addresses an important practical problem in engine health monitoring under genuinely difficult sensing conditions.
- The use of atomic-vapor-based spectral selectivity for Na/K-related emission is an interesting cross-disciplinary idea.
- The paper presents an integrated end-to-end pipeline from sensing to detection to 3D reconstruction, which is more compelling than an isolated component paper.

### Major Weaknesses

- The sensing mechanism is not explained with enough precision to support the central imaging claim. In Section 3.1.1, the paper appears to mix several distinct physical ideas, including narrowband atomic filtering, nonlinear frequency conversion, quantum jump / emission, and Faraday rotation, but it remains unclear which mechanism actually produces the final image contrast. As written, it is difficult to determine the image-formation model: what photons are emitted, what photons are filtered, and what role the atomic vapor cell plays beyond a high-level "suppression" description. This ambiguity is not just a presentation issue; it directly affects the credibility of the downstream detection and reconstruction results, because the sensing module is the foundation of the full pipeline.

- The assumed link from Na/K spectral response to actual wear particles is not directly validated. Appendix B provides useful spectral evidence for Na/K-related peaks, but this does not show that the bright clusters in the images are true wear particles rather than other Na/K-bearing combustion products, droplets, or background artifacts. The paper needs a more direct correspondence experiment between detected image regions and independently verified particles.

- The detection evaluation is not transparent about ground truth construction. The detection section reports very high mAP / recall while also stating that true particle counts or exact ground truth are extremely difficult to obtain. This creates a central evaluation gap: how were boxes or counts labeled, and against what reference were they verified? Without a clear annotation and verification protocol, the reported accuracy is hard to interpret.

- The current train/validation split may substantially overestimate detection performance. The paper mentions training on large numbers of frames extracted from continuous recordings, which are highly correlated in time. If the split is frame-level rather than by clip, engine run, or recording session, temporal leakage could substantially inflate the reported mAP and recall.

- The single-camera 3D reconstruction claim is under-justified. The paper argues that temporal particle rotation provides effective multi-view information, but it does not clearly explain how relative viewpoints or poses are established in real jet flow from a fixed camera. In Section 3.3.2, temporal silhouettes are treated as geometrically related views, but the paper does not provide a clear motion model, pose estimation procedure, or validation that this assumption holds in practice.

- The voxel fusion rule described in Section 3.3.2 appears geometrically problematic. The reconstruction section appears to describe a fusion rule closer to union / logical OR over silhouette-supported voxels. If this interpretation is correct, it seems inconsistent with standard silhouette / visual-hull-style reconstruction, which usually removes voxels that are inconsistent across views rather than preserving voxels supported by any single silhouette.

### Minor Weaknesses

- The empirical comparison set is narrower than desirable. More concrete baselines would help, such as classical thresholding / blob detection on the filtered images and a matched non-atomic optical baseline.
- The 3D evaluation would be stronger with more geometric metrics beyond image-style similarity measures such as SSIM, e.g., IoU, Chamfer distance, or F-score against known shapes.
- The paper should discuss deployment details and operating limits more concretely, including calibration procedure, sensitivity to fuel/additive composition, and failure modes when Na/K adhesion is weak or particle load is low.

---

> ### Author Rebuttal · Authors · 2026-03-30
>
> We thank R1 for the thorough and constructive review. We appreciate the recognition of our cross-disciplinary approach and integrated pipeline. We address each concern below:
>
> Q1: Image-formation mechanism. Thank you for identifying this critical ambiguity. We clarify this image-formation model as four steps: (1) the optical lens collects broad-spectrum light from the jet flow, which includes Na and K spectral signatures from wear particles; (2) the light enters a heated atomic vapor cell, within which gaseous Na/ K atoms absorb and re-emit photons at their specific transition energies, which purely fall in the spectrum of Na/K; (3) The cell is placed in an external magnetic field between customized polarizers, which rotates the polarization angle of the light passing through the atomic medium. (4) Finally, the polarizers block the unrotated, broad-spectrum background flame while transmitting only the rotated Na/K light. This makes the background flame invisible and enables high-contrast imaging of the wear particles, allowing the electro-optical imaging system to capture high-contrast images exclusively for the elemental Na/K on the wear particles. We will clarify this pipeline in the revised manuscript.
>
> Q2: Particle validation evidence. Thanks for the valuable suggestion. Because our system filters all light except Na/K, we lack direct optical evidence of wear particles and rely on indirect physicochemical and spatial evidence instead. At high temperatures, Na/K transfer electrons to the aluminum oxide in wear particles, causing strong adhesion. Consequently, bound Na/K forms dense clusters with clear trajectories, unlike unbound combustion products that move randomly and dispersedly. Combining these distinct behavioral patterns with spectral data allows us to confidently identify the dense, moving bright clusters as true wear particles.
>
> Q3: Ground truth construction. Thanks for the comment and sorry for the ambiguity. We used X-AnyLabeling, a state-of-the-art automatic labeling tool, to generate ground-truth bounding boxes for training our YOLOv8 model. This automated approach is highly reliable for our specific scenario since our optical system filters out background flame radiation, making the wear particles appear as distinct bright spots against a dark background. More importantly, because we use an ultra-high-speed camera (i.e., resolution 1280×800, pixel size 25.6 µm), consecutive frames are highly similar, highlighting only the movement of the particles. Even sub-pixel scale wear particles retain sufficient spatial detail to be identified and labeled without ambiguity. Therefore, X-AnyLabeling produces highly precise annotations that serve as robust ground truth with only minimal manual correction. We will add how boxes or counts were labeled in the revised manuscript.
>
> Q4: Train/validation split and temporal leakage. Thanks for the comments and sorry for the ambiguity. In our work, we use a strict, temporally independent split during YOLOv8 model training to avoid temporal leakage. 1) In fact, we split an 8min (480s) video into first 336s and last 144s for the same engine. 2) For different engines, we perform the split based on separate physical tests. Specifically, the model is trained entirely on data from one engine runs, i.e., continuous 7min and validated on a completely distinct, independent engine run, i.e., continuous 3min. We will modify the manuscript to clarify this ambiguity.
>
> Q5: 3D reconstruction pose estimation. Thanks for the comments. The key assumption underlying our approach is that the wear particles experience sufficient and continuous rotation within the camera's field of view. Following the Jeffery Orbits model, extreme shear rates in high-speed jet flows induce particle angular velocities reaching thousands of rad/s. This rapid rotation naturally exposes various facets of the irregular particles, providing the required angular diversity for 3D synthesis from 2D inputs. To clarify this, our revised manuscript will include relevant citations and a figure illustrating how these relative viewpoints are established.
>
> Q6: Voxel fusion rule. Thanks for catching this mistake. We agree that a logical OR of extruded silhouettes would indeed over-expand the volume. However, in extremely harsh environments like jet engines, a strict logical AND would permanently carve away parts of the particle if a single view is flawed. Therefore, we correct our method by adopting a “majority vote” strategy: a voxel is kept if it appears in at least half of the views. Otherwise, it is removed from the 3D shape. This hybrid strategy effectively prevents both the massive volume bloat of an OR and the catastrophic volume loss of an AND. We will modify those contents in the revised manuscript.
>
> Minor:
> All minor weaknesses have been addressed and will be added in the revised manuscript.

---

### Decision · Program_Chairs · 2026-04-30

**Decision:**

Accept (regular)

**Comment:**

This paper proposes QuantWear, a hardware-software system for sub-millimeter wear particle detection and 3D profiling in jet-engine exhaust, combining a quantum/atomic sensing module for Na/K-based signal isolation with a computer vision pipeline for particle enumeration and silhouette-based reconstruction. The reviews suggest that this paper addresses an important and technically challenging problem in jet-engine health monitoring, and that its cross-disciplinary integration of sensing and vision is a notable strength. The main concerns raised in the initial reviews include the clarity of the sensing mechanism and the validation of particle correspondence, the transparency of the evaluation protocol, the limited availability of real-world ground truth, and the technical description of the YOLOv8-based detection and reconstruction pipeline. The rebuttal provides substantially clearer explanations of the image-formation process, the modified YOLOv8 design, the frame-wise detection setting, and the voxel fusion strategy. Two reviewers explicitly state that their primary concerns have been resolved. At the same time, some clarifications still need to be incorporated more explicitly into the final manuscript, and parts of the evaluation remain reliant on indirect or simulation-based validation. Considering the importance of the problem, the system-level originality of the work, and the substantial clarifications provided in the rebuttal regarding the main technical concerns, the AC thinks that the paper meets the acceptance standard of ICML.